# The Effects of Compressive Residual Stress on Properties of Kyanite-Coated Zirconia Toughened Alumina Ceramics

**DOI:** 10.3390/ma16176071

**Published:** 2023-09-04

**Authors:** Hao-Long Wu, Haiyan Li, Dake Cao, Yan Qiu, Detian Wan, Yiwang Bao

**Affiliations:** 1China Building Material Test & Certification Group Co., Ltd., Beijing 100024, China; whl1079068629@126.com (H.-L.W.); lihaiyan@ctc.ac.cn (H.L.); caodake@ctc.ac.cn (D.C.); qiuyan@ctc.ac.cn (Y.Q.); dtwan@ctc.ac.cn (D.W.); 2State Key Laboratory of Green Building Materials, China Building Materials Academy, Beijing 100024, China

**Keywords:** kyanite, ZTA, flexural strength, residual stress, in situ synthesis

## Abstract

In this study, the prestressed coating reinforcement method was employed to create kyanite-coated zirconia toughened alumina (ZTA) prestressed ceramics. Due to the mismatch of the coefficient of thermal expansion (CTE) between the coating and substrate, compressive residual stress was introduced in the coating. The effects of compressive residual stress on the mechanical properties of ZTA have been demonstrated. Results show that the flexural strength of the kyanite-coated ZTA ceramics improved by 40% at room temperature compared to ZTA ceramics. In addition, the temperature dependence of mechanical properties has also been discussed. And the results show that the reinforcement gradually diminished with increasing temperature and eventually disappeared at 1000 °C. The modulus of elasticity of the material also exhibits a decreasing trend. Furthermore, the introduction of the prestressing coating enhanced the thermal shock resistance, but the strengthening effect diminished as the temperature increased and completely disappeared at 800 °C.

## 1. Introduction

Alumina ceramics are renowned for their outstanding mechanical properties and chemical stability, making them extremely well-suited for a wide range of industrial applications, whether at room temperature or at high temperatures. However, the industrial application range of alumina ceramics is limited by their relatively low fracture toughness [1,2]. In order to enhance their strength, researchers have investigated the utilization of zirconia-toughened alumina ceramics (ZTA). Zirconia-toughened alumina ceramics exhibit excellent mechanical properties and chemical stability. ZTA ceramics obtain relatively good fracture toughness compared to pure alumina ceramics, so they are applicable for use in various application fields, such as aerospace and refractories [3,4,5]. However, ZTA ceramics are still highly sensitive to defects. It is significant to develop ZTA ceramic components with high strength and exceptional damage tolerance.

In recent years, researchers have proposed a straightforward method for strengthening ZTA ceramics by applying a prestressed coating to the substrate. It has been reported that the flexural strength of alumina-coated zirconia has been enhanced to 1330 ± 52 MPa, which is 45% higher than that of pure zirconia ceramic [6]. Moreover, this method has been successfully applied in building ceramics, resulting in an increase in flexural strength from 44 ± 3 MPa to 89 ± 3 MPa [7].

Based on the design criteria of prestress coating [8], the kyanite-coated ZTA ceramics were obtained after pressure-less sintering. Previous studies have shown that kyanite can decompose into mullite and silica at temperatures above 1000 °C. The silica then reacts with the alumina on the composite’s surface to form mullite. Mullite is a SiO_2_-Al_2_O_3_ system, with the aluminum oxide content in the composition of mullite jumping between 72% and 78%. The sintering temperature is similar to that of alumina, allowing the coating material and substrate to be completely sintered at the same temperature, thus avoiding excessive or incomplete sintering. Additionally, the coefficient of thermal expansion of the kyanite-coated ZTA composite is lower than that of the ZTA substrate. This difference in thermal expansion coefficients leads to the generation of residual stress at the bonding interface during the cooling process. Furthermore, the temperature dependence of flexural strength and the residual stress of the kyanite-coated ceramics were also investigated.

## 2. Experiments

### 2.1. Materials

Kyanite powder (350 mesh), polyvinyl butyral (M.W. 9000–120,000, MACKLIN), castor oil (AR, MACKLIN), and alcohol (Jingchun reagent) were mixed in a ratio of 1:5%:0.5%:4 to prepare a coating slurry. After mixing, the weight ratio of the ball to the powder is 5:1. The prepared slurry was fully mixed by a planetary mill (YXQM-1L, MITR) for 5 h at a speed of 300 r min^−1^.

For preparing the ZTA substrate, 25 g of ZTA powder was dry pressed by using a stainless-steel mold with a diameter of 50 mm, and then the powder was compacted with a pressure of 300 MPa by using the cold isostatic pressing method, holding for 5 min. The compacted ZTA discs were pre-sintered at 1000 °C for 60 min, with a heating rate of 2 °C min^−1^, and kept at 600 °C for 1 h to remove the adhesive. Finally, we machined the pre-sintered ZTA disc into a rectangular shape with dimensions of 3.5 mm × 4.5 mm × 37 mm.

The pre-sintered ZTA substrates were soaked into a coating slurry, and the previous step was repeated when the desired coating thickness was achieved. The coating thickness is around 40~60 μm. Plain ZTA substrates were also prepared, and the ZTA substrate was sintered with the same sintering process as the coated samples. The prestressed composite structure and stress distribution are shown in Figure 1.

### 2.2. Characterization

A universal testing machine (Model C45, MTS, Akron, OH, USA) was used to test the three-point flexural strength of the ZTA substrate and prestressed composite, and the span and loading speed are 30 mm and 0.5 mm min^−1^, respectively. The data for each specimen were averaged over five tests. And, the section morphology was observed with a scanning electron microscope (Quanta 250FEG, FEI Company, Hillsboro, OR, USA). The Vickers indentation tester (Tukon2500B, Wilson, Norwood, MA, USA) produced indentation by setting appropriate loads (500 g or 1000 g) and holding time (15 s). The elastic modulus of the substrate and the prestressed composite was measured with RFDA-HT1600 (IMCE, Genk, Belgium), and then the elastic modulus of the kyanite coating was calculated using the formula mentioned in ISO20343 [9]. The coefficient of thermal expansion of the substrate and the prestressed composite was tested using a thermal expansion tester (Linseis, L75 platinum series, Selb, Germany), and then the coefficient of thermal expansion of the coating was calculated according to Equation (1).

The flexural strength of composite ceramics is effectively enhanced below 1000 °C. However, the prestressed reinforcement effect disappears due to stress relaxation as the testing temperature exceeds 1000 °C. To prove the above predictions, the relative method was used to calculate the corresponding residual compressive stress of the coating and residual tensile stress in the substrate at different test temperatures. The calculation method is as shown in Equations (1) and (2).
(1)$$\sigma_{c}=(\frac{S_{s}}{S_{c}})\cdot \left\{1-\left[\frac{E_{s}S_{s}}{E_{c}S_{c}}+\frac{\alpha_{c}}{\alpha_{s}}\right]/\left[1+\frac{E_{s}S_{s}}{E_{c}S_{c}}\right]\right\}E_{s}\cdot \alpha_{s}\cdot ∆T$$
where $\sigma_{c}$ is the residual stress of the coating; $S_{s}$ is the cross-sectional area of the substrate; $S_{c}$ is the cross-sectional area of the coating; $E_{c}$ is the elastic modulus of the coating; $E_{s}$ is the elastic modulus of the matrix; $\alpha_{c}$ is the coefficient of thermal expansion of the coating; $\alpha_{s}$ is the substrate coefficient of thermal expansion; and ∆T is the difference between the brittle-ductile transition temperature (zero stress temperature) and the ambient temperature. The residual compressive stress in the coating forms an equilibrium system with the residual tensile stress inside the substrate, where S represents the respective cross-sectional area.

The residual compressive stress in the coating forms an equilibrium system with the residual tensile stress inside the substrate, so the equation can be derived, as shown below.
(2)$$\sigma_{c}S_{c}=\sigma_{s}S_{c}$$
where $\sigma_{s}$ is the residual stress in the substrate.

To further evaluate the impact of residual stress on the strength of the ceramic substrate, the residual stress at different temperatures is calculated. This calculation was complemented by analyzing the high-temperature flexural strength test curve and considering previous studies on the temperature at which coating stress failure occurs. In [10], proposed by Bao, in the prestressed composite, the stress failure temperature is approximately 1000 °C, which corresponds to the minuend of the temperature drop range in Formula (1). The coefficient of thermal expansion of the composite and ZTA substrate samples is measured using a thermal dilatometer. The coefficient of thermal expansion of the coating material is calculated based on the principle of the relative method, using Formula (3). Additionally, the elastic modulus of the coating is calculated using the relative method, based on the elastic modulus of the two samples measured by the impulse excitation of vibration method, as described in Equation (4) [9,11].
(3)$$\alpha_{c}=\bar{\alpha }-\frac{E_{s}h_{s}}{E_{c}h_{c}}(\alpha_{s}-\bar{\alpha })$$
(4)$$E_{c}=\frac{{\left(1+2R\right)}^{3}E_{f}-E_{s}}{8R^{3}+12R^{2}+6R}$$
where $\bar{\alpha }$, $\alpha_{c}$, and $\alpha_{c}$ are the coefficient of thermal expansion of prestressed composite, the coefficient of thermal expansion of coating, and the coefficient of thermal expansion of substrate, respectively; $E_{f}$, $E_{c}$, and $E_{s}$ are the elastic modulus of the prestressed composite, the elastic modulus of the coating, and the elastic modulus of the matrix, respectively; $h_{c}$, $h_{s}$, and R are the coating thickness, substrate thickness, and the ratio of coating thickness to substrate thickness, respectively.

## 3. Results and Discussion

### 3.1. Microstructure and X-ray Diffraction

The prestressed composite was formed through pressureless sintering at 1600 °C, and an X-ray diffraction test was conducted on the prestressed coating. The X-ray diffraction pattern, shown in Figure 2, reveals the kyanite phase in the coating material, which disappeared after sintering. The mullite phase can be found in the XRD pattern. This can be attributed to the decomposition of the kyanite into mullite phase and SiO_2_ at 1500 °C. The synthesis of mullite occurred through the reaction between SiO_2_ and alumina in the coating slurry, as described by Equations (5) and (6). The results of XRD indicate that after pressureless sintering at 1600 °C for 2 h, the kyanite phase in the coating was almost completely transformed into the mullite phase, which agrees with the results from other researchers [12,13].
(5)$$3(Al_{2}O_{3}\cdot {\mathrm{SiO}}_{2})\to 3Al_{2}O_{3}\cdot 2{\mathrm{SiO}}_{2}+{\mathrm{SiO}}_{2}$$
(6)$$3Al_{2}O_{3}+2{\mathrm{SiO}}_{2}\to 3Al_{2}O_{3}\cdot 2{\mathrm{SiO}}_{2}$$

Figure 3 presents the cross-sectional morphology of the kyanite-coated ZTA ceramics. The figure clearly illustrates that the coating is tightly bonded to the substrate, indicating good interface compatibility between the kyanite coating and the ZTA substrate. This strong interface formation is attributed to the tensile stress calculated in Table 1, which confirms the tight bonding of the coating to the substrate interface.

### 3.2. Mechanical Properties

This experiment investigates the flexural strength and thermal shock resistance of ZTA ceramics with and without coatings. ZTA ceramics and kyanite-coated ZTA ceramics exhibit a decrease in flexural strength as the experimental temperature increases, as shown in Figure 4. When the temperature is below 1000 °C, the flexural strength of kyanite-coated ZTA composite ceramics is higher than that of ZTA ceramics, particularly at room temperature (25 °C). The flexural strength of kyanite-coated ZTA ceramics reaches 868 ± 24 MPa at room temperature, which is 40.0% higher than that of ZTA ceramics, at 620 ± 20 MPa. When the test temperature exceeds 1000 °C, the flexural strength of ZTA ceramics and kyanite-coated ZTA ceramics becomes approximately equal, so the use temperature of the kyanite-coated ZTA ceramics should be lower than 1000 °C. However, the intrinsic flexural strength of the mullite phase formed by the kyanite coating after sintering at 1600 °C is only 118 MPa [14]. During the cooling and shrinking processes, the coating on the ZTA substrate generates residual compressive stress due to the difference in thermal expansion coefficients. This residual compressive stress plays a crucial role in resisting internal and external tensile stresses, effectively inhibiting the propagation and extension of internal microcracks and material surface cracks. As a result, it increases the failure strain of the material [15]. After pressureless sintering, a prestressed composite with significantly higher strength than the individual materials was obtained. When the test temperature exceeds 1000 °C, the flexural strength of ZTA ceramics and kyanite-coated ZTA ceramics becomes approximately equal, suggesting that the application temperature of kyanite ZTA ceramics should be lower than 1000 °C. As depicted in Figure 5, the residual flexural strength of both materials decreases with increasing quenching temperatures. However, it is observed that the residual flexural strength of ZTA at 300 °C and kyanite-coated ZTA at 400 °C shows a steep fall. The critical temperature, at which the strength sharply drops off after quenching, is widely recognized as a crucial indicator for evaluating thermal shock resistance [13], but at the same quenching temperature, the strength of kyanite-coated ZTA is consistently greater than that of ZTA. Combined with Figure 6, it shows that after quenching, the surface of the ZTA produces a large number of small irregular cracks, and even some of the cracks have been extended; however, on the surface of the kyanite-coated ZTA, small cracks formed after quenching, and did not expand or connect with each other. The experimental results demonstrate that the residual compressive stress of the coating can partially balance the tensile stress induced by the external environment, thereby inhibiting defect extension and improving its strength. Additionally, the residual stress can effectively withstand the thermal shock experienced by the materials. In summary, the presence of residual compressive stress significantly enhances the flexural strength and thermal shock resistance of ZTA ceramics.

To investigate the strengthening mechanism of residual compressive stress, Vickers indentation experiments were conducted on ZTA ceramics and kyanite-coated ZTA ceramics. Figure 7a shows the indentation at the interface of ZTA, where the crack uniformly extends along the four corners of the indentation. This suggests that under uniform and no external pressure conditions, there is no compressive or tensile stress within ZTA. Figure 7b displays the cross-section of the kyanite-coated ZTA. Indentation near the edge of the sample is depicted in Figure 7d, while Figure 7c shows the indentation in the substrate. It is noticeable that the cracks perpendicular to the substrate coating interface are hindered, which is not observed in Figure 7d. This indicates that the coating exerts compressive stress on the substrate. Consequently, there exists a tensile stress in the substrate to counteract the compressive stress from the coating. As a result, the growth of cracks intensifies in the direction perpendicular to the coating substrate interface ${(a}_{1}<c_{1})$, as observed in Figure 7c.

### 3.3. Analysis of Residual Compressive Stress

The elastic modulus of the ZTA substrate and the kyanite-coated ZTA at each temperature point measured by the pulse excitation method is shown in Figure 8. As the temperature increases from room temperature to 1200 °C, E_f_ and E_s_ decrease slightly because the chemical bond between atoms weakens with the increase in temperature [16]. The coefficient of thermal expansion of the ZTA substrate and the kyanite-coated ZTA at each temperature point is measured with a thermal dilatomet (Figure 9).

The coefficient of thermal expansion and elastic modulus of the coating material can be calculated using the mentioned formula. The calculated data can then be substituted into Formulas (3) and (4) to determine the residual stress in both the coating and substrate of the prestressed composite at each temperature point (Figure 10). Furthermore, Table 1 presents the parameters used for calculating the residual stress at each temperature point during the high-temperature flexural strength test.

**Table 1 materials-16-06071-t001:** Residual stress in coating and substrate.

*E_f_*/GPa	*E_s_*/GPa	*E_c_*/GPa	*R*	$\bar{\alpha }$ × 10^6^/K^−1^	*α_c_* × 10^6^/K^−1^	*α_s_* × 10^6^/K^−1^	Δ*T*/°C	*σ_c_*/MPa	*σ_s_*/MPa
309.78	330.69	206.95	0.067	7.14	1.63	7.37	700	798.62	53.24
302.98	325.14	194.24	0.067	7.94	3.67	8.11	400	331.64	22.11
293.57	318.48	171.88	0.067	8.27	3.82	8.43	200	152.87	10.19
285.19	310.84	160.13	0.067	8.42	4.34	8.56	100	65.28	4.35
281.55	306.92	157.86	0.067	8.59	4.51	8.73	0	0	0
274.11	298.35	155.85	0.067	8.74	4.72	8.88	−100	−62.35	−4.18
262.39	284.62	153.79	0.067	8.87	4.98	9.01	−200	−119.54	−7.97

The results in Figure 4 demonstrate that the flexural strength values of ZTA and the kyanite-coated ZTA are equal after reaching a temperature of 1000 °C. This suggests that the prestressed effect disappears at 1000 °C, resulting in residual tensile stress and residual compressive stress turning to 0 (Figure 10). The analysis diagram and table indicate that the coating has a residual compressive stress effect on the substrate. There is a balance between the tensile stress and the substrate, which suggests that the kyanite prestressed coating generates a surface residual compressive stress on the ZTA substrate due to the difference in coefficient of thermal expansion. The presence of compressive stress in the surface coating can effectively prevent crack initiation and propagation, thereby enhancing the strength and toughness of prestressed ceramics. The negligible residual tensile stress further confirms the strong and relatively strong flexural strength of the kyanite coating and the ZTA substrate.

## 4. Conclusions

The aim of this study is to enhance the strength of the substrate material by applying pre-stressing techniques using suitable coatings and substrate materials. Numerous researchers have conducted studies in this area. For this study, kyanite was selected as the coating material, and ZTA was chosen as the substrate material. The two materials were sintered together to create pre-stressing composites through in-situ synthesis. The final conclusions of the study are outlined below:By utilizing kyanite material with a low thermal expansion coefficient as a coating, the strength of ZTA ceramics experienced a significant improvement of 40% at room temperature. However, as the temperature increases, the strength gradually decreases until it reaches 1000 °C (known as the zero-stress temperature). This decrease can be attributed to the reduction of residual compressive stress in the surface layer as the temperature rises until it reaches 0 at 1000 °C.A comparison was made between the flexural strengths of coated and uncoated ZTA specimens at different temperatures. The results show that the zero-stress temperature of the kyanite-coated ZTA ceramics was approximately 1000 °C. Beyond this temperature, the strength of the kyanite-coated ZTA ceramics and ZTA ceramics became similar. This similarity can be attributed to the relaxation of compressive residual stress in the coating.The residual flexural strength of the quenched kyanite-coated ZTA is found to be higher than that of ZTA up to a temperature of 800 °C. This indicates that the presence of residual compressive stress in the coating improves the material’s resistance to thermal shock. This observation is consistent with the discovery that the quenched kyanite-coated ZTA shows significantly reduced surface crack extension compared to the ZTA.

## Figures and Tables

**Figure 1 materials-16-06071-f001:**
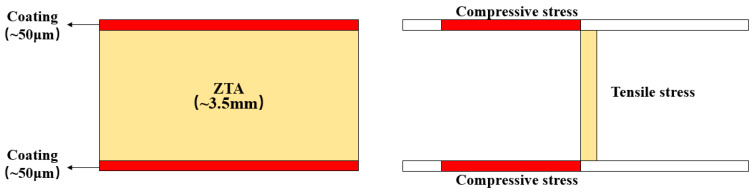
Prestressed composite structure and stress distribution.

**Figure 2 materials-16-06071-f002:**
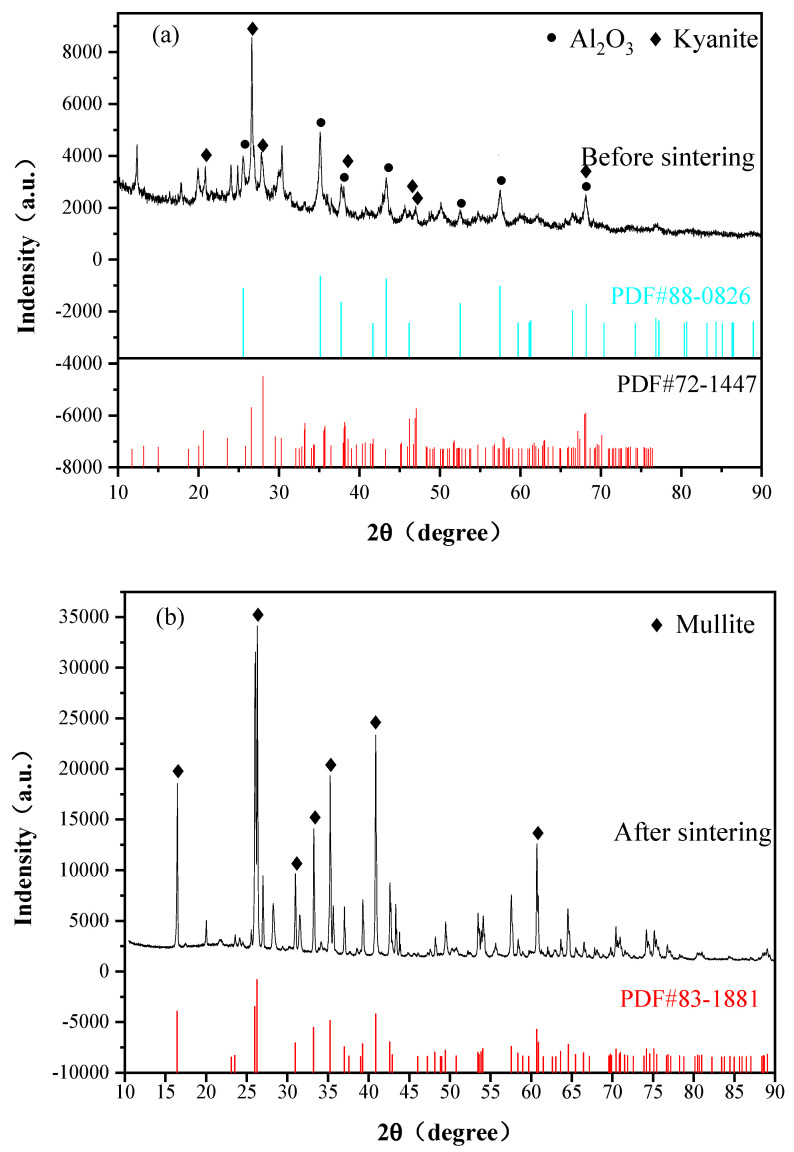
XRD patterns of the coating materials: (**a**) coating components before being sintered; (**b**) coating components after being sintered.

**Figure 3 materials-16-06071-f003:**
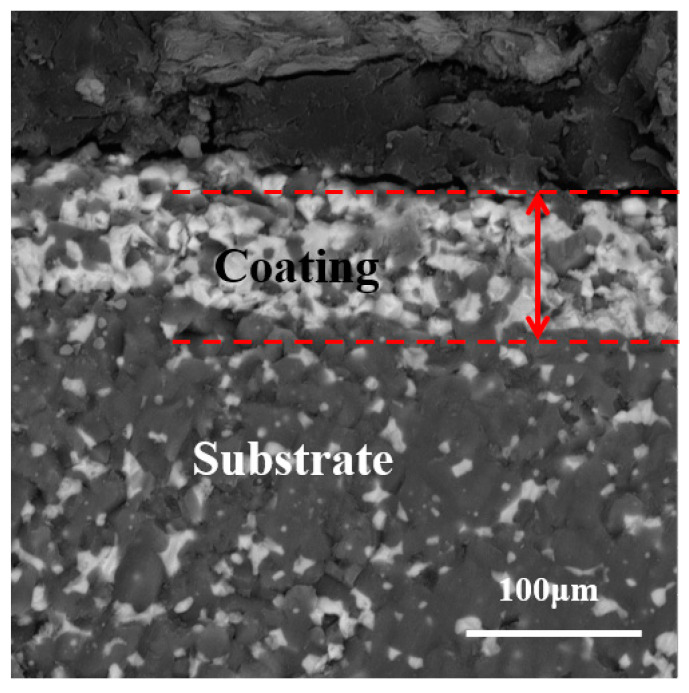
Cross-section morphology of prestressed composite materials.

**Figure 4 materials-16-06071-f004:**
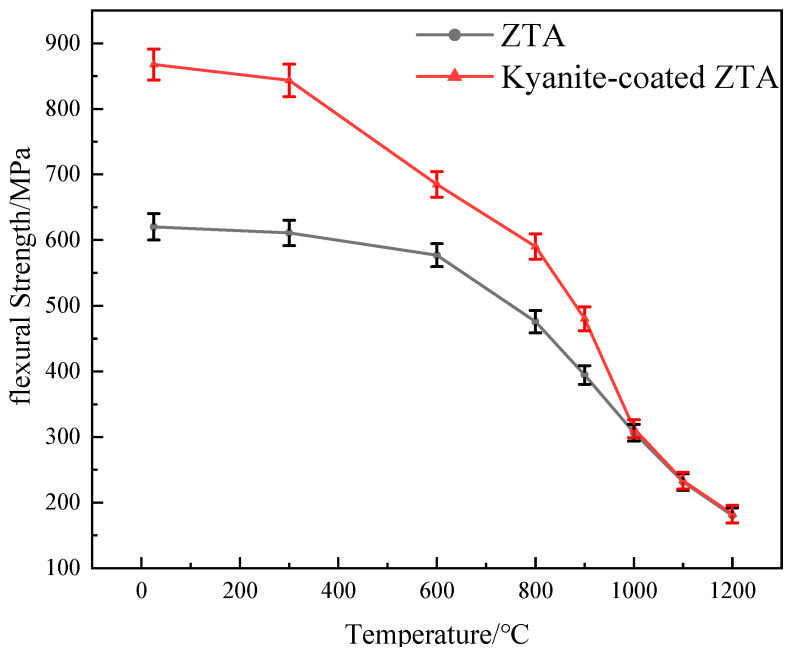
Temperature-dependent curves of the flexural strength of substrate and composite.

**Figure 5 materials-16-06071-f005:**
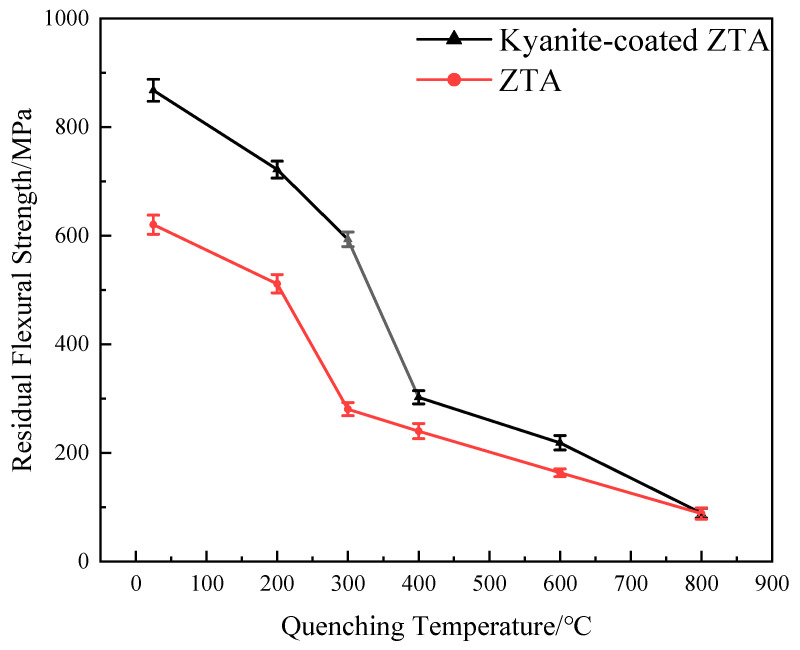
Residual flexural strength of substrate and composite measured after quenching at different temperatures.

**Figure 6 materials-16-06071-f006:**
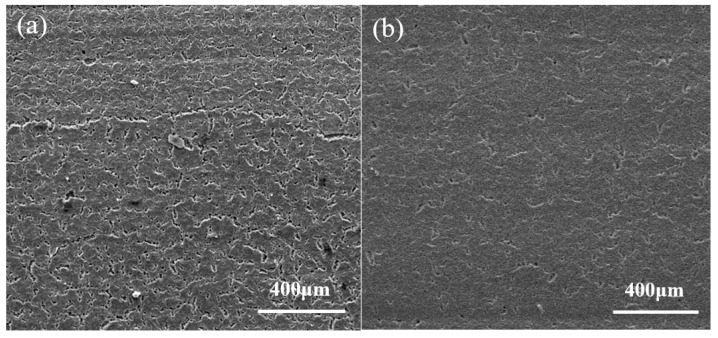
SEM images of material surfaces. (**a**) Surface morphology of ZTA after quenching, and (**b**) surface morphology of kyanite-coated ZTA after quenching.

**Figure 7 materials-16-06071-f007:**
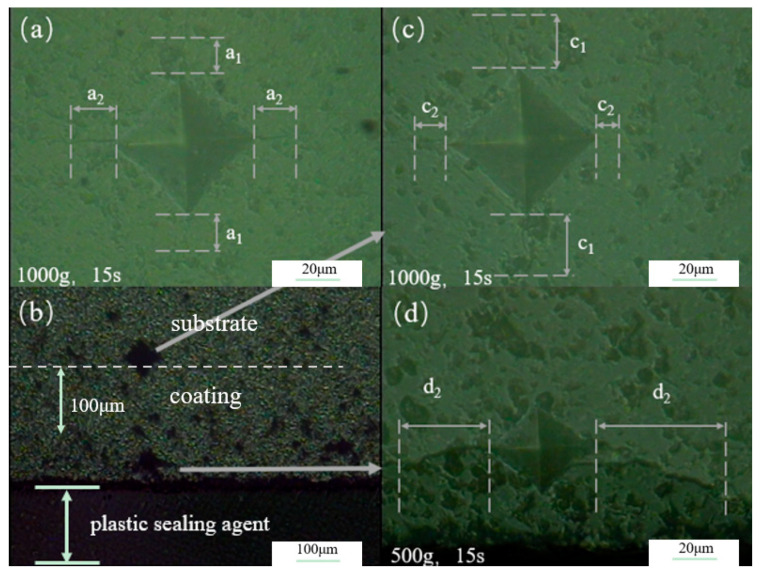
Composite interface indentation. (**a**) Interface indentation of the ZTA; (**b**) interface indentation of the kyanite-coated ZTA; (**c**) interface indentation on the substrate of the kyanite-coated ZTA; (**d**) interface indentation on the coating of the kyanite-coated ZTA.

**Figure 8 materials-16-06071-f008:**
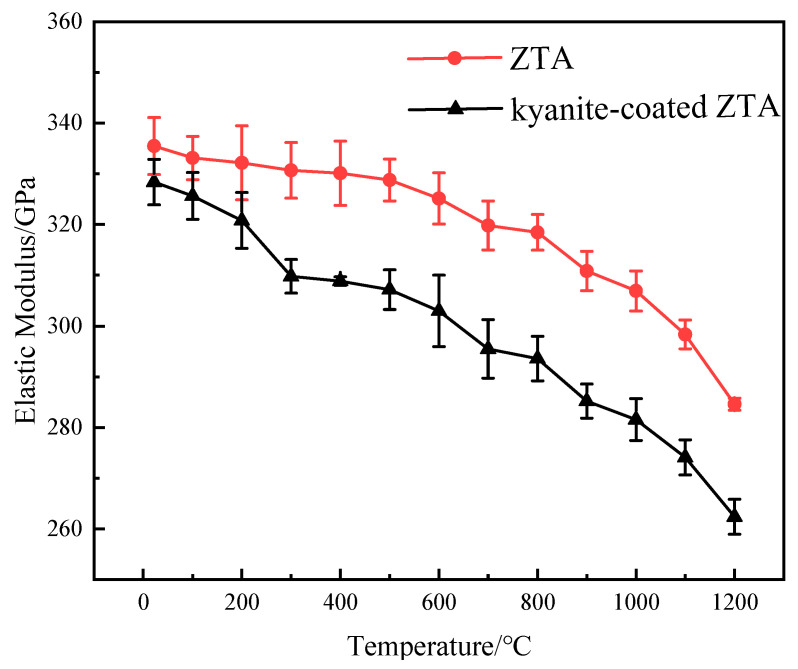
Temperature-dependent curves of the elastic modulus of substrate and composite.

**Figure 9 materials-16-06071-f009:**
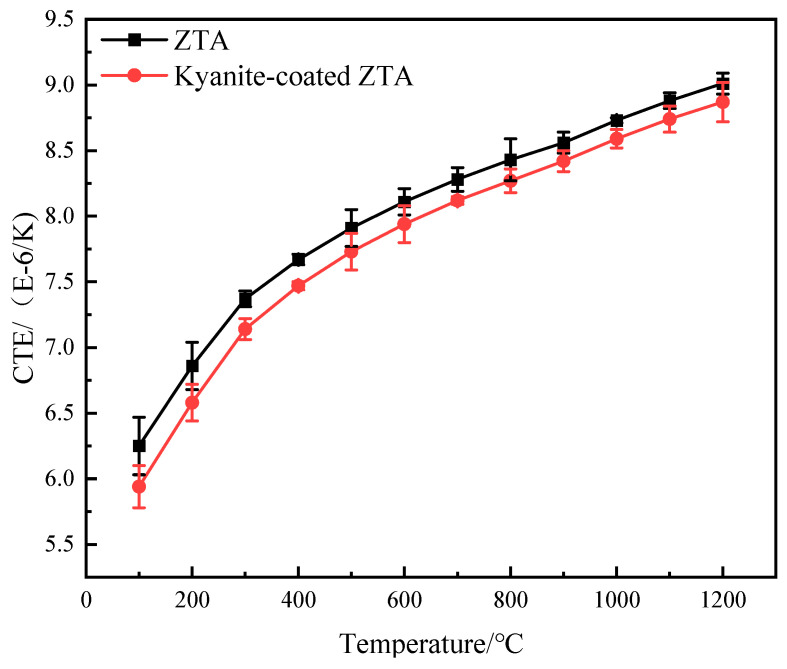
Curves of the coefficient of thermal expansion of the substrate and complex changing with temperature.

**Figure 10 materials-16-06071-f010:**
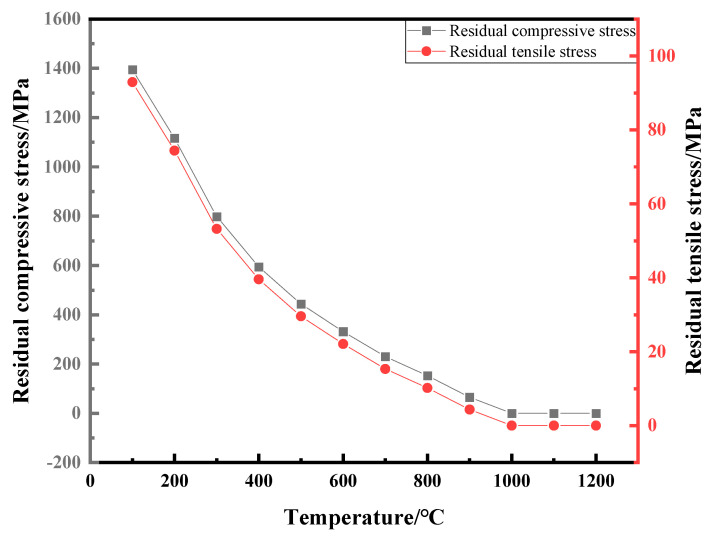
Temperature-dependent curves of residual stress in coating and substrate.

## Data Availability

Data supporting this study are included within the article.

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
