# Peer review of "The Effects of Compressive Residual Stress on Properties of Kyanite-Coated Zirconia Toughened Alumina Ceramics"

_materials, 2023, doi:10.3390/ma16176071_

Round 1

Reviewer 1 Report

Reviewer’s Comments:

The manuscript “The Effects of Compressive Residual Stress on Properties of Kyanite-Coated Zirconia Toughened Alumina Ceramics” is very interesting work. This paper investigates the prestressed coating reinforcement method was employed to create kyanite-coated zirconia toughened alumina (ZTA) prestressed ceramics in this study. Due to the mismatch of the coefficient of thermal expansion (CTE) between the coating and substrate, the compressive residual stress was introduced in the coating. The effects of compressive residual stress on the mechanical properties of ZTA have been demonstrated. Results show that the flexural strength of the kyanite-coated ZTA ceramics was improved by 40% at room temperature compared to ZTA ceramics. However, the following issues should be carefully treated before publication.

1. In abstract, the author should add more scientific findings.

2. Keywords: the synthesized system is missing in the keywords. So, modify the keywords.

3. In the introduction part, the introduction part is not well organized and cited references should cite the recently published articles such as 10.1016/j.colsurfa.2022.129332 and 10.1016/j.jphotochem.2021.113393

4. Introduction part is not impressive and systematic. In the introduction part, the authors should elaborate on the scientific issues in the Alumina Ceramics research.

5. Results …, The author should provide reason about this statement “Indentation near the edge of the sample is depicted in Figure 7(d), while Figure 7(c) shows the indentation in the substrate”.

6. The authors should explain regarding the recent literature why “proposed by Bao, In the prestressed composite, the stress failure temperature is approximately 1000 , which corresponds to the minuend of the temperature drop range in formula (3)”.

7. The author should explain the latest literature “Furthermore, Table 1 presents the parameters used for calculating the residual stress at each temperature point during the high-temperature flexural strength test”.

8. The author should provide reason about this statement “In addition, the presence of residual stresses in the kyanite coating inhibits cracking of the surface during quenching, thereby improving the material's resistance to thermal shock”.

9. Comparison of the present results with other similar findings in the literature should be discussed in more detail. This is necessary in order to place this work together with other work in the field and to give more credibility to the present results.

10. The conclusion part is very weak. Improve by adding the results of your studies.

Moderate editing of English language required

Reviewer 2 Report

This paper investigates the effects of a kyanite coating on the properties of zirconia toughened alumina (ZTA) ceramics. The study finds that the flexural strength of the kyanite-coated ceramics is significantly improved by 40% compared to non-coated ceramics at room temperature. The coating induces compressive residual stress on the substrate, which counteracts the tensile stress from the environment and inhibits crack propagation. This strengthening effect is most pronounced at temperatures below 1000°C, beyond which the residual stress decreases and the strength of coated and uncoated ceramics becomes similar. Additionally, the kyanite coating enhances the thermal shock resistance of the ceramics. The study was conducted by Hao-Long Wu, who performed material preparation, tests, data collection, and analysis. The research was funded by the National Natural Science Foundation of China, and there are no conflicts of interest. The article includes relevant references on the topic of alumina ceramics and their properties.

As a general comment, authors should more clearly state a hypothesis or, failing that, a main aim at the end of the introduction. The conclusions should refer to the achievement of this aim or confirm the demonstration of their hypothesis.

Other specific comments are as follows:

Line 27: prevent MPa from moving to the next sentence by using Ctrl+Shift+Space in MSWord.

Line 30: remove the brackets from the magnitude (applies to the whole paper).

Line 42: Put a period before "A ceramics" and a space.

Line 46: Put "Kynanite" and a space between "power(350.."

Line 48: Define "certain" as it is not very concrete.

Line 50: Remove a space between "a speed".

Line 50: Units should be expressed as r min^-1. Use this format throughout the rest of the document.

Line 54: Prevent units from moving to the next line.

Line 69: Put a space between "microscope(".

Line 70: type (500 g or 1000 g).

The expressions that appear in section 3.3 (lines 164 to 187) should appear in the experimental section after line 76, as the authors did not derive them and they are used by other authors.

Line 84: add SiO2 as subscript.

Figure 2. Replace the symbol ♥ with another one. Add the caption "XRD patterns of the coating materials: (a) before sintering and (b) after...".

Lines 109 and 110: use the correct significant figures: 868 ± 24 and 620 ± 20 and remove the brackets as indicated above.

Figure 4. The title of the ordinates would be "Bending".

Figure 5. Put the label of the figure on page 5/11.

Figure 6. Same comment for the caption of the figure as in Figure 2.

Line 157: add a space between "interface(".

Line 158: remove the space "c1, also...".

Figure 7. Correct the caption by inserting "," instead of ";", the necessary spaces and "and" in the last term of the enumeration.

Section 3.3. Move lines 164-187 to the experimental section, otherwise put "," between the numbers of expressions (3) and (4) (line 169).

Line 191: Insert "." instead of ";".

Line 193: delete "as shown in" and insert "(Figure 9)".

Figure 9: move the caption of the figure to page 8/11.

Line 202: prevent "10" from moving to the next line (use Ctrl+Shift+Space).

Line 208: change "Figure 4" to "Figure 10".

Conclusions: The first sentence of the conclusions should state whether the aim of the work has been achieved or the working hypothesis has been successfully tested. Avoid the use of bullet points.

References: Format the references according to the Materials format (read the instructions for authors) or use a software program such as EndNote or similar to format the references in the correct style.

Round 2

Reviewer 2 Report

I am grateful to the authors for making the changes I suggested in the review. In this new version, 2 minor changes need to be made:

Line 62: change "kyanite" to "kyanite".

Line 63: I think I didn't explain correctly, the units should be r [space] min [superscript] -1. I meant by "^" that it should be an exponent or that the letter should be a superscript. I hope this clarifies my point, the problem is that the plain text of the platform does not allow me to express it better. This comment applies to the rest of the document.

Regarding the conclusions, I accept the authors' correction, but I recommend for future work not to use enumeration, as it looks like an academic work and detracts from their great work, it is preferable to use paragraphs.
